# Analysis and optimization of fire evacuation safety performance in large urban complexes

**Yunhao Jiang**[1], **Gang Liu**[1]*, **Yulun Du**[1], **Siteng Cai**[1], **Zhichao Si**[1], **Jing He**[1], **Xiangbing Zhou**[2]*

**1** College of Geography and Planning, Chengdu University Of Technology, Chengdu,Sichuan, China,
**2** Key Laboratory of Mountain Tourism Safety, Sichuan Tourism University, Chengdu, Sichuan, China

* liugang2014@cdut.edu.cn (GL); zhouxb@uestc.edu.cn (XZ)

## Abstract

Urban large-scale complexes, such as shopping malls, pose significant challenges for fire safety management due to their intricate spatial layouts, high population density, and diverse occupancy characteristics. Efficient fire evacuation strategies are critical for minimizing casualties and economic losses; however, existing approaches often overlook the dynamic interplay between fire propagation and human behavior, resulting in suboptimal safety assessments. This study proposes an integrated simulation framework to optimize evacuation strategies by coupling fire dynamics with pedestrian flow modeling, aiming to enhance both evacuation efficiency and personnel safety. The methodology comprises three key steps: (1) Fire scenario simulation: A Building Information Modeling (BIM)-based digital platform is constructed to simulate fire propagation. Critical fire parameters (e.g., heat release rate, combustion model) are calibrated to quantify temporal variations in smoke temperature, CO concentration, and visibility across different zones. (2) Evacuation dynamics modeling: A pedestrian evacuation model is developed by integrating demographic factors (age structure, movement speed, population density) and fire-induced regional risks, enabling realistic simulation of crowd movement under fire conditions. (3) Safety performance evaluation and strategy optimization: Safety margins at staircases are assessed by comparing Required Safe Egress Time (RSET) and Available Safe Egress Time (ASET), followed by a safety grading system to identify high-risk bottlenecks. Evacuation strategies are then optimized to mitigate these risks. A case study was conducted on a shopping mall in Chengdu to validate the framework. Simulation results indicate an initial evacuation time of 260.4 seconds. Safety performance analysis revealed critical risks at staircases A and C (1st floor) and D (2nd floor) due to insufficient safety margins. After strategy optimization, the total evacuation time was reduced to 245.5 seconds, with safety margins at the three high-risk staircases increased by 130.8 s, 115.2 s, and 72 s, respectively, fully meeting safety requirements. The overall evacuation efficiency was significantly improved. This study

**Data availability statement:** The dataset can be found through Figshare at https://doi.org/10.6084/m9.figshare.29755559.v1.

**Funding:** Open Funding of Sichuan Provincial Key Laboratory of Philosophy and Social Sciences for Mountain Tourism Safety (Grant number 24SDLYAQZZ001), National Natural Science Foundation of China Project (Grant number 42371418).

demonstrates the effectiveness of the proposed framework in quantifying fire risks and optimizing evacuation strategies for large-scale complexes. The integrated simulation approach provides a scientific basis for evidence-based safety management and evacuation planning, offering valuable insights for urban fire safety engineering and emergency response optimization.

## Introduction

With the rapid socioeconomic development in China and the growth of commercial activities, the demand for shopping venues has shifted from traditional single-purpose spaces to multifunctional large-scale commercial complexes that integrate consumption, entertainment, office, and dining. However, due to their complex spatial structures, dense facilities, and high population density, safety risks in large commercial complexes are intricate and severe [1]. In recent years, several major fire incidents have occurred in large commercial complexes in China. For instance, in 2018, a fire caused by an electrical short circuit at the "Hao Yi Xin" Commercial City in Sichuan resulted in one fatality. In 2021, a fire caused by improper cutting operations at the Tongluowan Commercial Plaza in Anhui led to four deaths and two injuries. In 2022, a fire broke out at Jinsheng Department Store in Nanjing due to long-term use of an induction cooker, which ignited leftover materials in the pot. These incidents highlight the critical importance of conducting emergency evacuation simulations for large commercial complexes in the event of a fire.

Given the complexity of fire causes and processes in buildings, various factors influence personnel evacuation. Traditional experimental methods for studying fire evacuation are time-consuming, labor-intensive, and costly, making it difficult to obtain accurate fire simulation results [2].Building Information Modeling (BIM) is a digital technology-based tool for managing the entire lifecycle of a building. Its core lies in constructing a 3D model that incorporates the building's physical properties, functional characteristics, and full-process data, thereby achieving information integration and collaborative management throughout the building's stages from planning and design, construction, operation and maintenance to even demolition. In the field of fire protection, BIM models can accurately integrate detailed information of the building's fire protection system, providing precise geometric and attribute data input for fire simulation software and evacuation simulation tools. This ensures that the simulation scenarios are highly consistent with the actual situation of the building, thereby enhancing the scientificity of fire risk assessment and evacuation strategy optimization. Integrating BIM with Fire Dynamics Simulator (FDS) and EVAC technology allows for effective simulation of building fire evacuations, significantly reducing research costs and time, while providing a more intuitive representation of the evacuation process during a fire emergency [3]. Scholars at home and abroad have conducted extensive research on integrating BIM technology with building fire evacuation. For instance, using BIM as a foundation, virtual reality technology has been applied to simulate fire emergency evacuation in virtual environments [4]. FDS and EVAC have been employed to simulate fire scenarios and personnel evacuation,

enabling comparison between available safe evacuation time and required safe evacuation time to assess building safety and optimize existing evacuation routes [5]. An environment-aware beacon deployment algorithm combined with BIM technology has been developed to achieve real-time personnel tracking, enhancing the efficiency and accuracy of emergency response [6]. Wireless local area networks, ultra-wideband, and radio frequency identification technologies have been used to retrieve building data from BIM and generate wiring networks for indoor navigation [7]. FDS and EVAC were utilized to simulate emergency evacuation in a high-rise residential project, where the spread of smoke was analyzed based on fluid dynamics principles, and the safety of construction workers' emergency evacuation was evaluated [8]. A Cellular Automaton (CA) model was employed to simulate safe evacuation, analyzing the position of obstacles within the building and optimizing site layout based on the analysis results [9]. An agent-based model was used to simulate and analyze the human tolerance limits in complex commercial building fire environments, determining the available evacuation time for personnel [10]. A fire evacuation numerical analysis model was employed to study the effects of parameters such as evacuation population size, minimum group effect, and pre-movement time on evacuation performance, proving that a 50.0% increase in the evacuation population leads to crowd congestion, which becomes the critical factor determining evacuation time [11].

As can be seen from the above, currently, there are many studies on building fire evacuation using BIM technology in combination with Pyrosim and Pathfinder software. However, most of these studies only determine the fire evacuation safety performance of buildings based on simulation results, while there are relatively few studies that optimize fire prevention designs in combination with data. Due to their particularities, large urban complexes have a relatively high fire risk. Therefore, it is necessary to analyze the fire evacuation safety performance of large urban complexes and carry out research on fire hazard prevention measures and the optimization of fire prevention designs.

This study constructs a BIM-based digital platform to dynamically couple fire dynamics simulation with pedestrian flow modeling, thereby realizing the realistic scenario simulation of crowd evacuation under fire conditions and establishing a dynamically coupled multi-dimensional simulation system. It proposes a quantitative assessment system for building fire evacuation safety performance based on safety margins. By guiding the pedestrian flow to staircases with high safety margins, targeted optimization of evacuation strategies is achieved. Taking a shopping mall in Chengdu as an example and combining real data, fire evacuation simulations are conducted and personnel evacuation plans are optimized, which verifies the effectiveness of the proposed framework. This research provides a scientific basis for shopping mall safety management, fire protection design and optimization, and emergency plan formulation, filling the gap in existing studies where there is a lack of data-integrated optimization for fire protection designs.

## Scenario construction

In this study, a shopping center in Chengdu was selected as the research scenario for personnel evacuation under fire conditions. The shopping center is a four-story frame structure with a total floor area of 11,108 m², with each floor covering 2,777 m² and a floor height of 4 meters. The first floor has seven exits connected to the outside and seven staircases. The shopping center includes various commercial establishments such as leisure tea houses, food courts, and department stores, featuring complete facilities, high customer flow, dense occupancy, and numerous combustible materials.

The BIM model of the shopping center was constructed using an architectural template. Grid lines were drawn based on building drawings, with floor elevations and slab thicknesses configured. Wall models were built according to the drawings, with wall dimensions configured and the wall type set as "basic wall." Windows, doors, and staircases were then added. The constructed BIM model was used to create the fire simulation model and evacuation model, as shown in Fig 1.

## Fire simulation

**Fire simulation parameter settings.** Thunderhead Engineering Pyrosim (Pyrosim Version 2021) is a modeling software specifically designed for Fire Dynamics Simulation (FDS). Pyrosim 2021 is used to simulate and predict the flow

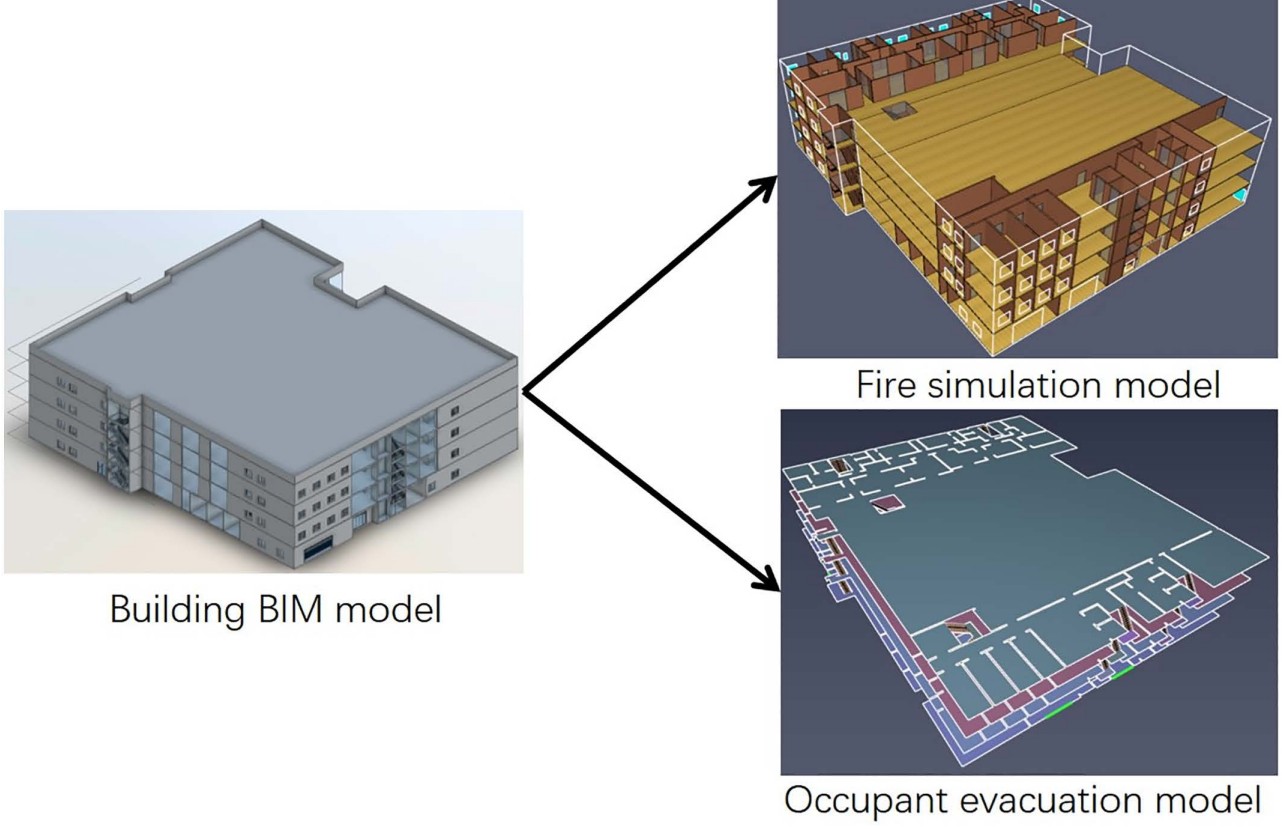

**Fig 1. BIM based fire evacuation model for shopping centers.**

of smoke, carbon monoxide, and other toxic gases, as well as fire temperatures and smoke concentration distribution during fires.

Pathfinder (Pathfinder Version 2021) is an evacuation simulation software that models escape processes in different building environments by setting parameters such as occupant numbers, walking speeds, distances, and paths, thereby evaluating building evacuation designs and safety.

This study employs Pyrosim and Pathfinder for fire simulation and evacuation analysis. When designing fire scenarios, key considerations typically include the specific location of the fire, the quantity of combustible materials, occupant distribution, and the structural characteristics of the building. When setting up fire scenarios, the most likely ignition points should be identified based on the actual conditions of the large-scale complex [12].

The subject building is a large shopping center with a substantial floor area. Based on practical analysis, the restaurant on the first floor is identified as the most probable fire origin. Restaurant kitchens frequently use liquefied petroleum gas (LPG) and natural gas for open-flame cooking, which can lead to oil fume accumulation and potential ignition. Additionally, the concentration of various electrical equipment increases the risk of short-circuit fires. Located in the central area of the shopping center, smoke from the first-floor restaurant can quickly spread through windows and doors to the main hall and adjacent rooms. As the fire develops, smoke will fill stairwells and, under high temperatures, rapidly spread to the second and third floors, making this a high-risk and reasonable fire scenario setup.

Fire combustion simulations can be conducted using steady-state or unsteady-state models. While steady-state fire calculations are relatively simpler, real-world fires are typically unsteady-state, with the heat release rate (HRR) varying

over time. The HRR is proportional to the square of time. Therefore, this study adopted the unsteady-state fire model, expressed as:

$$Q = \alpha t^2 \tag{1}$$

Where: Q is the heat release rate of the fire source (kW);α is the fire power growth coefficient (kW/s²); t is the fire duration (s).According to the four standard fire types defined in NFPA 204−2021 "Standard for Smoke and Heat Venting" [13] and China's "Code for Fire Protection in Building Design" GB 50016−2014 [14].

This simulation assumed a fire scenario where a short circuit ignites combustible polyurethane materials in the restaurant.The fire's heat release rate is set at 10 MW, classified as a fast-growth fire with a corresponding growth coefficient of 0.0469 kW/s². The fire area is 4 m², with a unit area heat release rate of 2500 kW/m². The time to reach the maximum heat release rate is 230.88 s. Under the worst-case fire conditions, the smoke yield and CO yield are set at 0.131 g/g and 0.1 g/g, respectively.

The heat release rate growth coefficient depends on the type of combustible material, and different types of combustibles can lead to distinct types of fires. Shopping malls contain a large variety of combustible materials, such as wood, PVC, and foam materials that are widely used in decoration processes. In this paper, the types of combustible materials are set as wood, PVC, and foam materials, while the walls are set as concrete. The material parameters are presented in Table 1.

When conducting fire simulations using the BIM model, grid division is necessary. Computational grids are a critical component of fire models, significantly impacting model establishment and calculation speed. Due to the irregular shape of the model, a multiple-grid approach is adopted to balance computational efficiency and accuracy, with four grids covering the calculation area.

Grid sensitivity analysis indicates that the empirical value for grid size should be 1/4–1/16 of the characteristic flame diameter [15]. The characteristic flame diameter D* is calculated as follows:

$$D^* = \left( \frac{\dot{Q}}{\rho_\infty C_p T_\infty \sqrt{g}} \right)^{\frac{2}{5}} \tag{2}$$

Where: $\dot{Q}$ is the heat release rate of the fire source (kW); $\rho_\infty$ is the air density (1.2 kg/m³); $C_p$ is the specific heat capacity of air (1 kJ/(kg·K)); $T_\infty$ is the ambient air temperature (293 K); g is gravitational acceleration (9.81 m/s²).

Based on the characteristic diameter calculation and considering the building's actual size and computational accuracy requirements, the grid size is determined to be 0.2 m × 0.2 m × 0.2 m, totaling 5,596,668 grids. Verification confirms that this grid size meets the requirement of D*/5.

To verify grid independence, the following criteria are typically required:

**Table 1. Material properties and parameters.**

| Material Type | Density | Specific Heat | Conductivity | Emissivity | Absorption Coefficient |
|---|---|---|---|---|---|
| Wood | 640 kg/m³ | 2.85 kJ/(kg·K) | 0.14 W/(m·K) | 0.9 | 5.0E+4 1/m |
| PVC | 1380 kg/m³ | 1.44 kJ/(kg·K) | 0.16 W/(m·K) | 0.95 | 5.0E+4 1/m |
| Foam | 28 kg/m³ | 1.7 kJ/(kg·K) | 0.05 W/(m·K) | 0.9 | 5.0E+4 1/m |
| Concrete | 2280 kg/m³ | 1.04 kJ/(kg·K) | 1.8 W/(m·K) | 0.9 | 5.0E+4 1/m |

1. Grid Convergence: When calculations are performed using different grids, the results should converge to similar solutions under sufficiently small grid sizes. As the grid size decreases, the computational results should gradually approach a stable solution.

2. Grid Independence: When calculations are conducted using different grids, the results should remain consistent under the same physical conditions. Even with varying grid sizes, the computational results should exhibit the same physical trends and characteristics.

3. Grid Smoothness: When calculations are carried out using different grids, the results should remain smooth under the same physical conditions. Even as the grid size changes, the computational results should not display significant isolated or singular points.

One-way Analysis of Variance (One-Way ANOVA) is a statistical method used to determine whether there are significant differences in the means among multiple groups. Its core principle involves decomposing the total variation into between-group variation and within-group variation, and then using the F-test and P-value to judge the significance of between-group differences. This method is well-suited for the research design of this study, where "grid scale serves as the sole independent variable and key indicators act as continuous dependent variables". It complies with the assumptions of independence, normality, and homogeneity of variance for grid monitoring data, enabling quantitative verification of whether grid scale exerts a significant impact on key indicators and thus accurately aligning with the objective of verifying grid independence.

Therefore, this study set three groups of gradually reduced grid sizes (1.2m × 1.2m × 1.2m, 0.6m × 0.6m × 0.6m, and 0.2m × 0.2m × 0.2m). Taking visibility (V), CO concentration (CO), and temperature (T) near the fire source as the key observation indicators, a one-way ANOVA was conducted based on the continuous monitoring data obtained 600 seconds after the fire ignition (with a sample size of 1001 for each group). The differences in the variation curves of the same fire product yield under different grid scales were compared to verify grid independence. The variation of fire product yields under different grid sizes is illustrated in Fig 2, and the results of the one-way ANOVA are presented in Table 2.

As shown in Fig 2, the variation trends of different fire product yields with fire development are similar under different grids and there are no significant divergence points. As presented in Table 2, in the results of one – way analysis of variance, among the yields of each fire product under different grid scales, the F – statistics of all indicators are relatively small, and all P – values are greater than 0.05, which meets the homogeneity of variance test. This indicates that changes in grid size will not significantly alter the results of key indicators such as visibility, CO concentration, and temperature, verifying that grid independence holds.

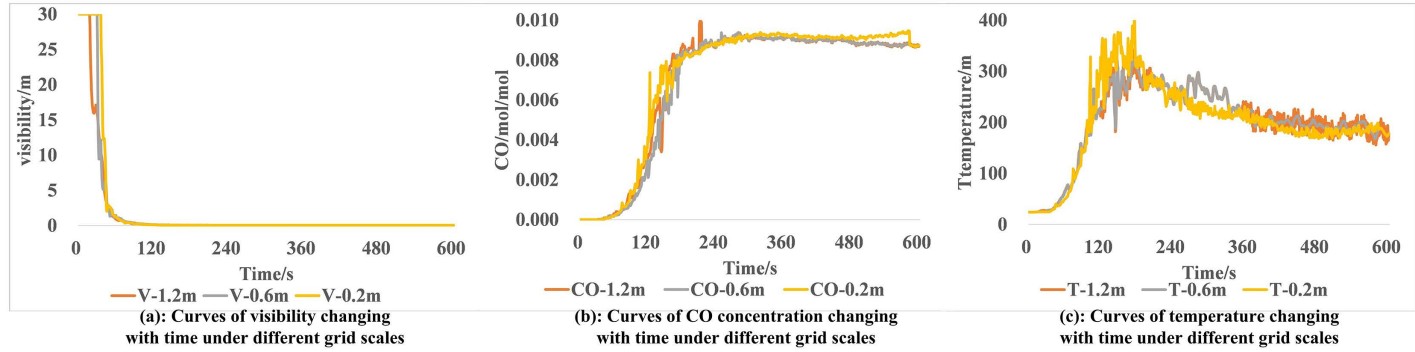

(a): Curves of visibility changing with time under different grid scales

(b): Curves of CO concentration changing with time under different grid scales

(c): Curves of temperature changing with time under different grid scales

**Fig 2. Variation of fire product yields under different grid sizes.**

**Table 2. One-way ANOVA results table of key indicators under different grid scales.**

| Analytical Indicato | Grid Scale | Group Mean (± Standard Deviation) | F-statistic | P-value | Levene's Test | Conclusion (α=0.05) |
|---|---|---|---|---|---|---|
| Visibility (V) | 1.2m | 10.23±12.65 | 1.555 | 0.211 | F=1.124 P=0.326 | Homogeneous variance; No significant difference between groups |
| | 0.6m | 10.87±12.71 | | | | |
| | 0.2m | 11.02±12.83 | | | | |
| CO Concentration(CO) | 1.2m | 0.008±0.003 | 1.624 | 0.197 | F=0.987 P=0.373 | Homogeneous variance; No significant difference between groups |
| | 0.6m | 0.007±0.002 | | | | |
| | 0.2m | 0.007±0.002 | | | | |
| Temperature (T) | 1.2m | 125.68±89.32 | 0.482 | 0.618 | F=1.352 P=0.259 | Homogeneous variance; No significant difference between groups |
| | 0.6m | 126.15±88.97 | | | | |
| | 0.2m | 127.03±89.15 | | | | |

During evacuation under fire conditions, quantitative calculations of smoke layer height are necessary to minimize its impact on occupants. The smoke layer height is calculated as:

$$H_S \geq H_C = H_P + 0.1 H_B \tag{3}$$

Where: $H_S$ is the smoke height (m); $H_C$ is the critical hazard height (m); $H_P$ is the average height of personnel (m); $H_B$ is the building's internal height (m).

According to the "Report on Chinese Residents' Nutrition and Chronic Disease Status (2020)" [16], the average height of Chinese males and females aged 18–44 is 1.697 m and 1.58 m, respectively. Adopting a conservative approach, we set the average male height at 1.7 m. With a building interior height of 4 m, the critical hazardous height of the smoke layer is calculated to be 2.1 m. Therefore, CO concentration, smoke temperature, and visibility slices are set at 2.1 m on each floor to monitor fire development across different planes and track changes in CO concentration, smoke temperature, and visibility over time.

Visibility and CO concentration detectors are installed at key evacuation nodes, such as major corridor turns and stairwells on each floor, to collect data and establish a digital smoke model. The building's internal layout is shown in Fig 3.

### Analysis of fire simulation results

**Smoke temperature variation analysis.** Fig 4 illustrates the temperature changes across different floors of the building. At 161 s after fire ignition, the smoke temperature in the first-floor restaurant reached 60°C. By the end of the simulation (600 s), only areas such as stairwells E and G on the first floor approached 60°C, while temperatures on other floors remained below this critical threshold.

As time elapses after ignition, due to the narrow space and limited airflow in the room where the fire source is located, heat accumulates more easily within the space, resulting in significant temperature changes. In contrast, in other areas, being relatively far from the fire source, and after fire smoke enters the hall, the internal space of the hall is larger, heat exchange is frequent, and the smoke dissipates more easily. These areas are significantly affected by the attenuation of thermal radiation and smoke energy transfer, leading to smaller temperature variations, all of which remain below 60°C.

**CO concentration variation analysis.** The variation of CO concentration on each floor is shown in Fig 5. After the fire occurred, the CO concentration inside the restaurant reached $1 \times 10^{-3}$ mol/mol at 192.4 seconds. By the end of the simulation at 600 seconds, only some areas in the first-floor lobby had a CO concentration approaching $1 \times 10^{-3}$ mol/mol, resulting in a relatively small affected area.

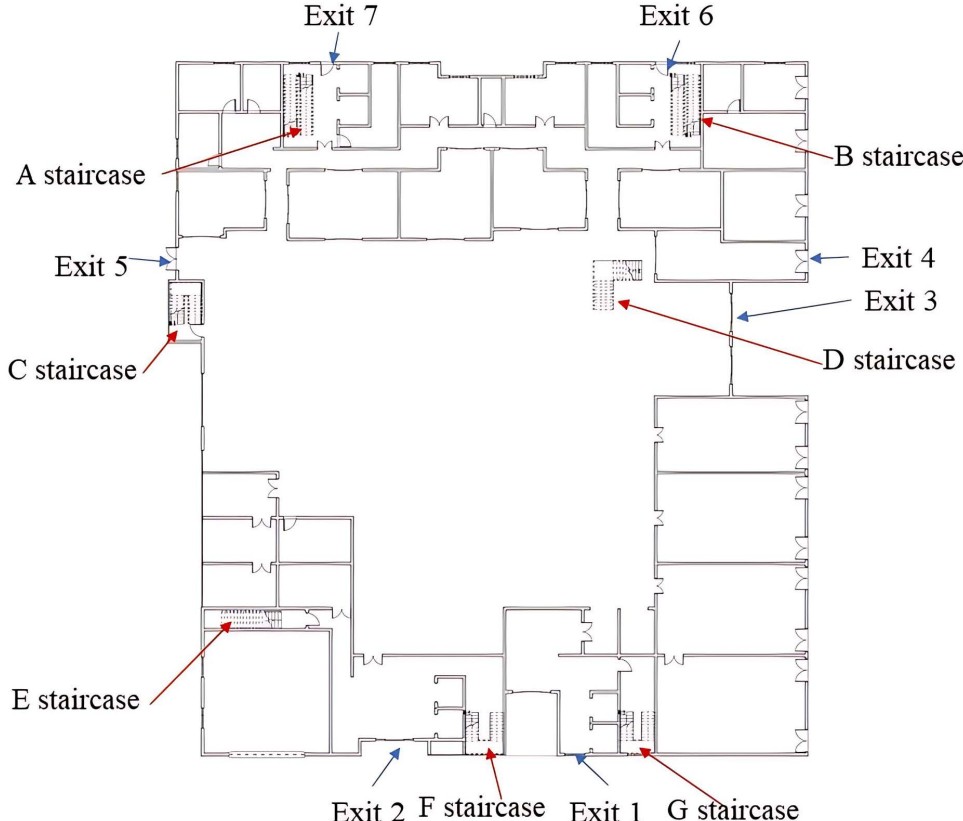

**Fig 3. Distribution of fire sources and staircases in a shopping center.**

**Visibility variation analysis.** Visibility is generally defined as the maximum distance at which a person can clearly see an object. The visibility conditions on each floor are illustrated in Fig 6. When a fire breaks out, a large amount of smoke is generated and quickly spreads to the surrounding rooms. By 142.5 seconds, the visibility in the first-floor restaurant dropped below 10 meters, exhibiting obvious horizontal diffusion characteristics. After 374.7 seconds of the fire, the smoke spread to the vicinity of staircases C and D, and the vertical upward spread of the smoke caused the visibility on the second floor to gradually drop below 10 meters, with the smoke continuing to spread to upper floors. After 571.3 seconds of the fire, due to the deposition of smoke particles under the action of gravity, the visibility in most areas of the first and second floors continues to decrease. The corridor areas where staircases A and B are located have narrow spaces and poor ventilation conditions, leading to severe smoke accumulation and a rapid drop in visibility to below 10 meters. As for staircases C and D, their closer distance to the fire source results in longer smoke accumulation time, causing their visibility to drop below 10 meters. At the end of the 600-second simulation, the visibility in most positions on the first and second floors was below 10 meters, with only the areas near staircases E and G being less affected in terms of visibility. On the third floor, the visibility at staircases A, B, C, and the lobby near staircase D drops below 10 meters, while the visibility in other areas remains good.

As mentioned earlier, the evacuation time should be determined by the earliest time when environmental conditions become life-threatening to evacuees. By comparing the impact scope and timing of smoke temperature, CO concentration, and visibility changes during the fire simulation, it is evident that visibility changes occur more rapidly and affect a

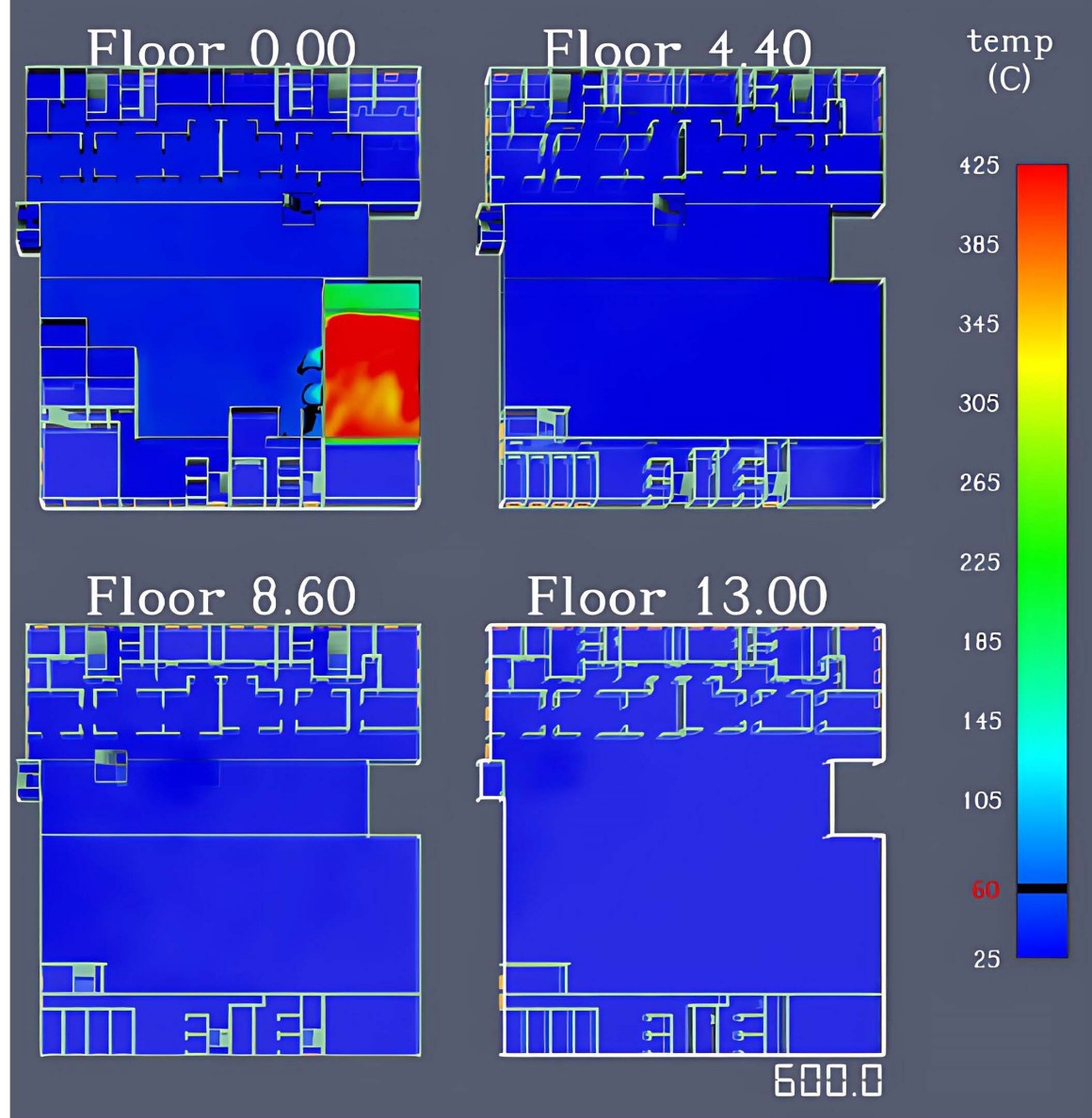

**Fig 4. Simulation results of smoke temperature variation across floors in the shopping center.**

wider area compared to smoke temperature and CO concentration changes. Therefore, the safe evacuation time for occupants is primarily determined by smoke visibility.

## Personnel evacuation

**Evacuation simulation parameter settings.** Personnel evacuation movement modes include: the SFPE mode, which ignores mutual interference between individuals during evacuation, and the Steering Mode [17],which accounts for collisions between evacuees and the spatial environment. Considering that the movement of emergency evacuation

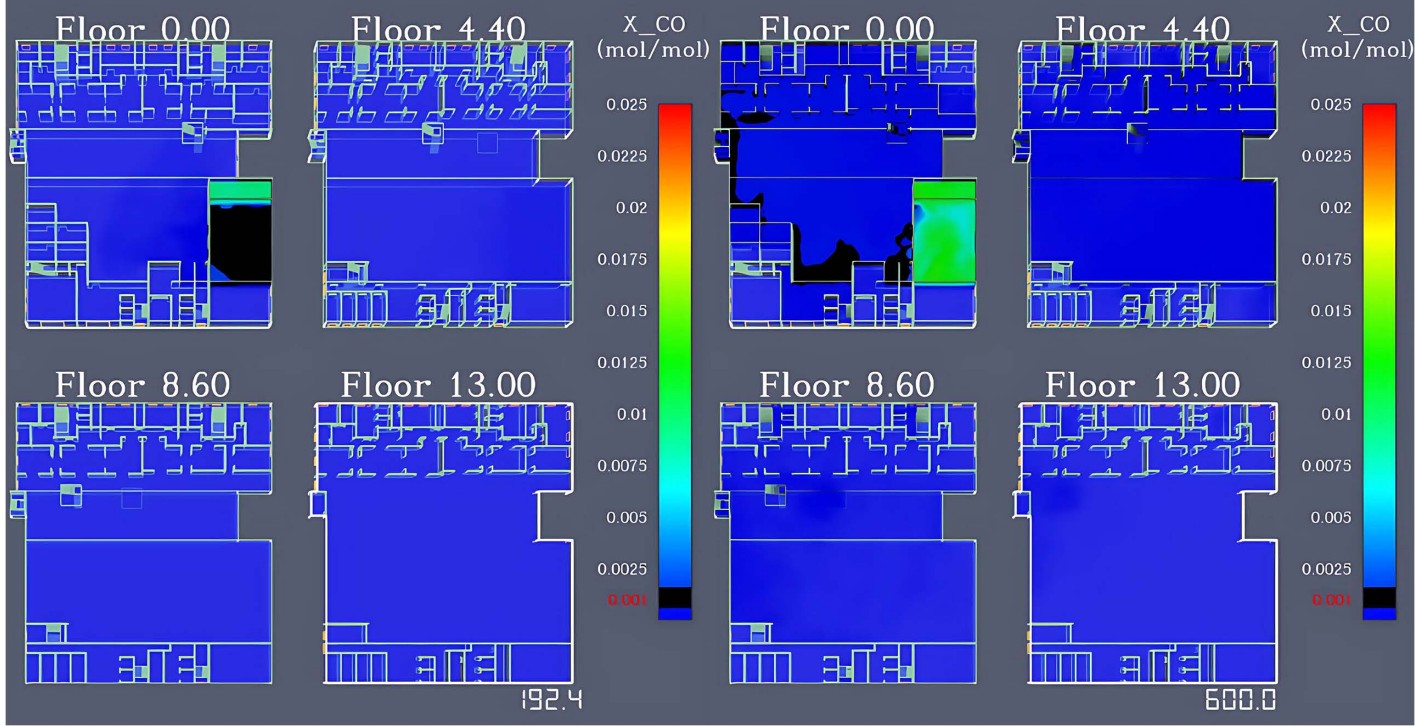

**Fig 5. Simulation results of CO concentration variation across floors in the shopping center.**

personnel in a fire will be subject to more interference factors, the Steering mode, which is closer to reality, was adopted for simulation in this experiment [18].

According to the Bulletin of the Seventh National Population Census of Chengdu, among the permanent population of Chengdu, the proportion of male population is 50.26%, and that of female population is 49.74%. The proportion of the population aged 0–14 is 13.28%, the proportion of the population aged 15–59 is 68.74%, and the proportion of the population aged 60 and above is 17.98%. Considering differences in physical attributes such as height, shoulder width, and walking speed that affect evacuation speed among people of different genders and ages, the evacuees are divided into four categories: children, adult males, adult females, and the elderly, with a ratio of 13.28%, 34.55%, 34.19%, and 17.98% respectively. The number of people on the first to fourth floors is set as 504, 424, 374, and 213 respectively. According to materials such as the Report on Nutrition and Chronic Diseases of Chinese Residents (2020), Fire Safety Engineering Part 9: Guidelines for Evacuation Assessment GB/T31593.9 - 2015 [19], and Anthropometric Dimensions of Chinese Adults (GB10000–2023) [20], the walking speeds and shoulder widths of different populations are shown in Table 3.

## Evacuation simulation results analysis

The change of the number of people on each floor with time is shown in Fig 7. When the fire had been burning for 20 s, the crowd started to move. When the fire had been burning for 134 s, the personnel on the fourth floor entered the stairwell and evacuated to the next floor through the stairs. When the fire had been burning for 209 s, the personnel on the third floor entered the stairwell and evacuated to the next floor through the stairs. When the fire had been burning for 239 s, the personnel on the second floor entered the stairwell and evacuated to the next floor through the stairs. Until 260.4 s, all personnel have been evacuated.

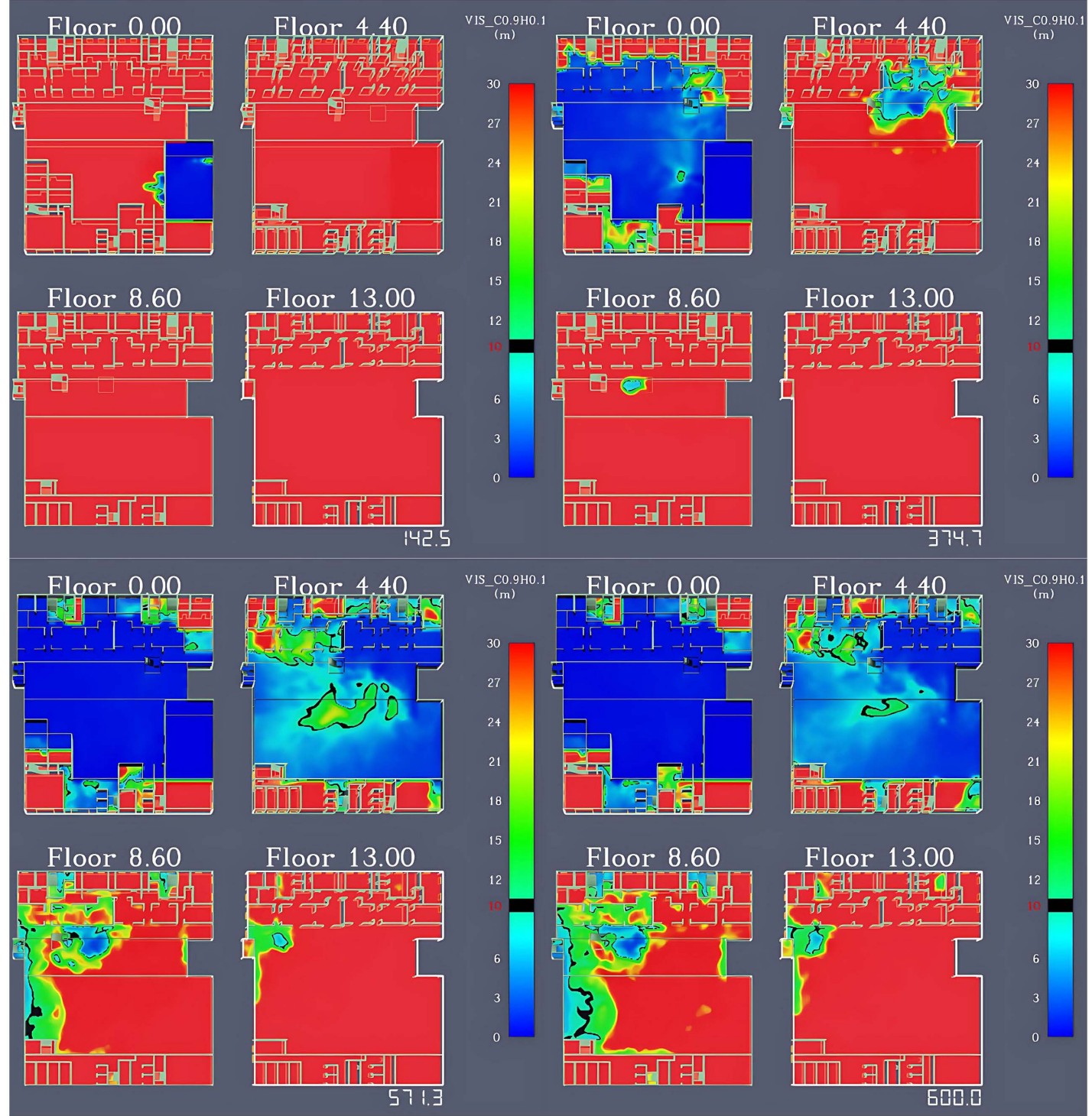

**Fig 6. Simulation results of visibility variation across floors in the shopping center.**

**Table 3. Parameters of Evacuees in a Shopping Mall.**

| Personnel type | Walking speed (m/s) | Shoulder width (m) | Height (m) |
|---|---|---|---|
| Adult male | 1.32 | 0.416 | 1.738 |
| Adult female | 1.29 | 0.378 | 1.61 |
| Children | 1.27 | 0.325 | 1.464 |
| Old people | 1.1 | 0.395 | 1.537 |

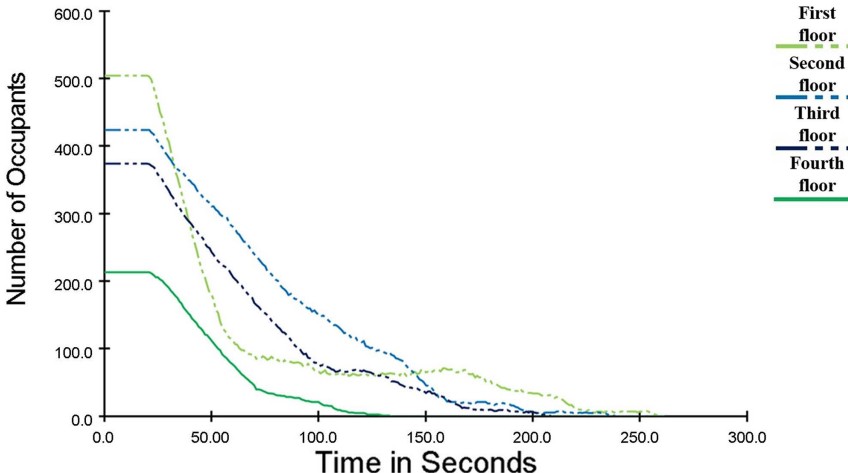

**Fig 7. Curve of the Number of Evacuees on Each Floor of a Shopping Mall Changing with Time.**

The relationship between evacuation time and the change in the number of people is shown in Fig 8. After 100 s of evacuation, half of the crowd had completed evacuation. The evacuation time for the remaining half was approximately 160 s, indicating a significant reduction in their evacuation speed compared to the first half.

The variation of pedestrian flow rate at each exit with evacuation time is shown in Fig 9. During the evacuation process, people tended to choose the nearest exit as their target. At 50 seconds after the start of evacuation, the pedestrian flow rate at each exit reached its peak: the flow rate at Exit 1 is the highest, reaching 6.17 people/s; the flow rates at Exit 2 and Exit 5 reach 3 people/s; and the maximum flow rates at the remaining exits are around 2 people/s. Afterwards, the flow rates gradually decrease until 110 seconds into the evacuation, when the pedestrian flow rate at each exit stabilizes at approximately 1 people/s. As indicated in the table, without evacuation guidance, although the pedestrian flow rates at each exit stabilize after a certain period, the distribution of people is uneven and fails to avoid fire-affected areas, resulting in reduced evacuation efficiency and a narrowed safety margin for evacuation.

### Improvement strategies for safety margin enhancement

**Safety margin evaluation method.** After a fire occurs, the available safe evacuation time for the building occupants should consist of four parts: the time from the occurrence of the fire to its detection, the fire alarm time, the personnel evacuation time, and the safety margin [21]. Fig 10 shows the relationship among these four elements, and the calculation formula is as follows:

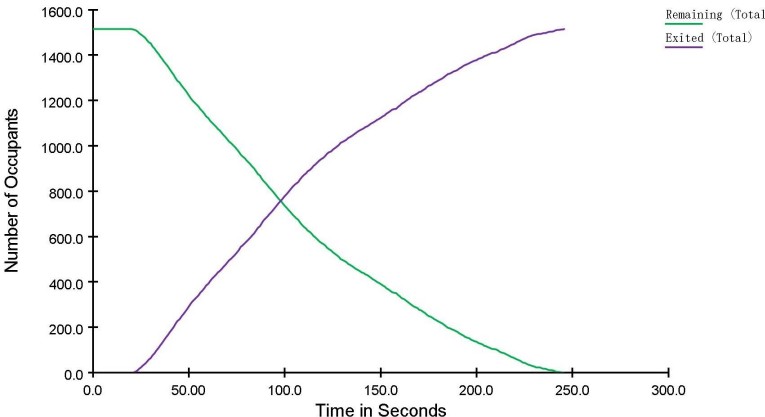

**Fig 8. Curve of the Number of Evacuated and Un – evacuated People Changing in a Shopping Mall.**

$$T_{rest} = T_{det} + T_{res} + \alpha T_{move} \qquad (4)$$

$$T_{aset} = T_{rest} + T_{surp} \qquad (5)$$

Where: $T_{aset}$ is the available safe evacuation time; $T_{det}$ is the time from the outbreak of the fire to its detection and alarm activation; $T_{res}$ is the time for people to hear the alarm and begin evacuation; α is a correction coefficient (set at 1.2); $T_{move}$ is the time required for personnel to evacuate; $T_{rset}$ is the required safe evacuation time; $T_{surp}$ is the safety margin.

Personnel Evacuation Time:As the most critical part of fire evacuation, several safety factors must be considered when calculating evacuation time. These include high-temperature burns caused by the heat of the fire, reduced visibility due to smoke, and dizziness, suffocation, or unconsciousness from inhaling toxic gases like CO [22].

Therefore, the critical thresholds for smoke temperature, CO concentration, and visibility that do not pose a threat to human life are used as evaluation standards:

1   Smoke Temperature: Studies show that second-degree burns can occur when humans are exposed to smoke temperatures of 71 °C for 60 seconds, 82 °C for 30 seconds, or 100 °C for 15 seconds. Based on data from the American National Standards Institute, 60 °C is selected as the critical danger threshold for fire smoke temperature.

2   CO Concentration: In the event of a fire, large amounts of smoke gather at the top of a room, and inhalation of CO and other toxic gases can severely affect evacuation [23] If people are exposed to air with a CO concentration exceeding $5 \times 10^{-4} 1 \times 10^{-3}$ mol/mol for more than 2 hours, they begin to feel discomfort; if exposed to air with a concentration of $1 \times 10^{-3}$ mol/mol for over 2 hours, they experience symptoms such as headaches and vomiting, which significantly hinder evacuation. Therefore, a CO concentration greater than $1 \times 10^{-3}$ mol/mol is chosen as the threshold.

3   Smoke Visibility: The quantitative standard for visibility should be determined by the height and area of the building space [24]. In general, visibility in large spaces should not be less than 10 meters, so the safe smoke visibility is set at 10 meters.

$T_1$ be the time when smoke temperature reaches 60° C, $T_2$ be the time when CO concentration reaches $1 \times 10^{-3}$ mol/mol, and $T_3$ be the time when smoke visibility decreases to 10 meters. To ensure personnel safety, the evacuation time is defined as:

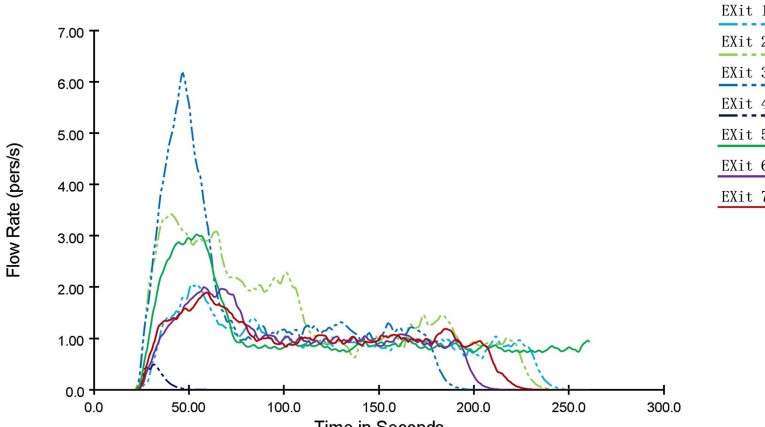

**Fig 9. Curve diagram of the variation of pedestrian flow rate with time at each exit of a shopping mall.**

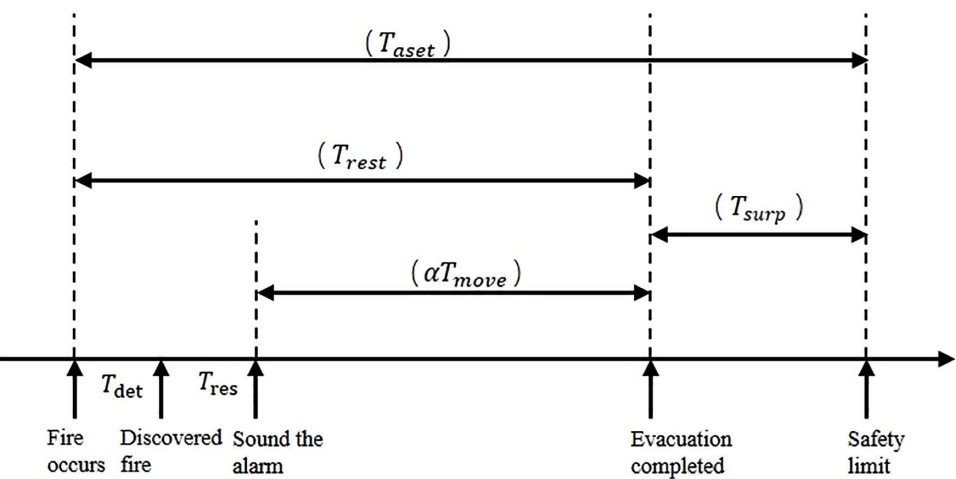

**Fig 10. Available safe evacuation time in a fire.**

$$T_{\text{move}} = \min(T_1, T_2, T_3) \tag{6}$$

Safety margin: During fire evacuation, an additional safety margin should be allocated beyond the required evacuation time to give occupants more time for preparation and to handle unexpected situations. This maximizes the likelihood that all personnel can safely evacuate, even under unforeseen circumstances [25]. The larger the safety margin in a fire evacuation, the higher the fire evacuation safety performance.

## Analysis of current safety performance

During the evacuation process, people are mainly affected by visibility. The changes in visibility at the stairway entrances on each floor are shown in Fig 11. The usage of each staircase during the evacuation process is shown in Fig 12. The available safe evacuation time is determined by the time it takes for the visibility at the stairway entrances to drop to the

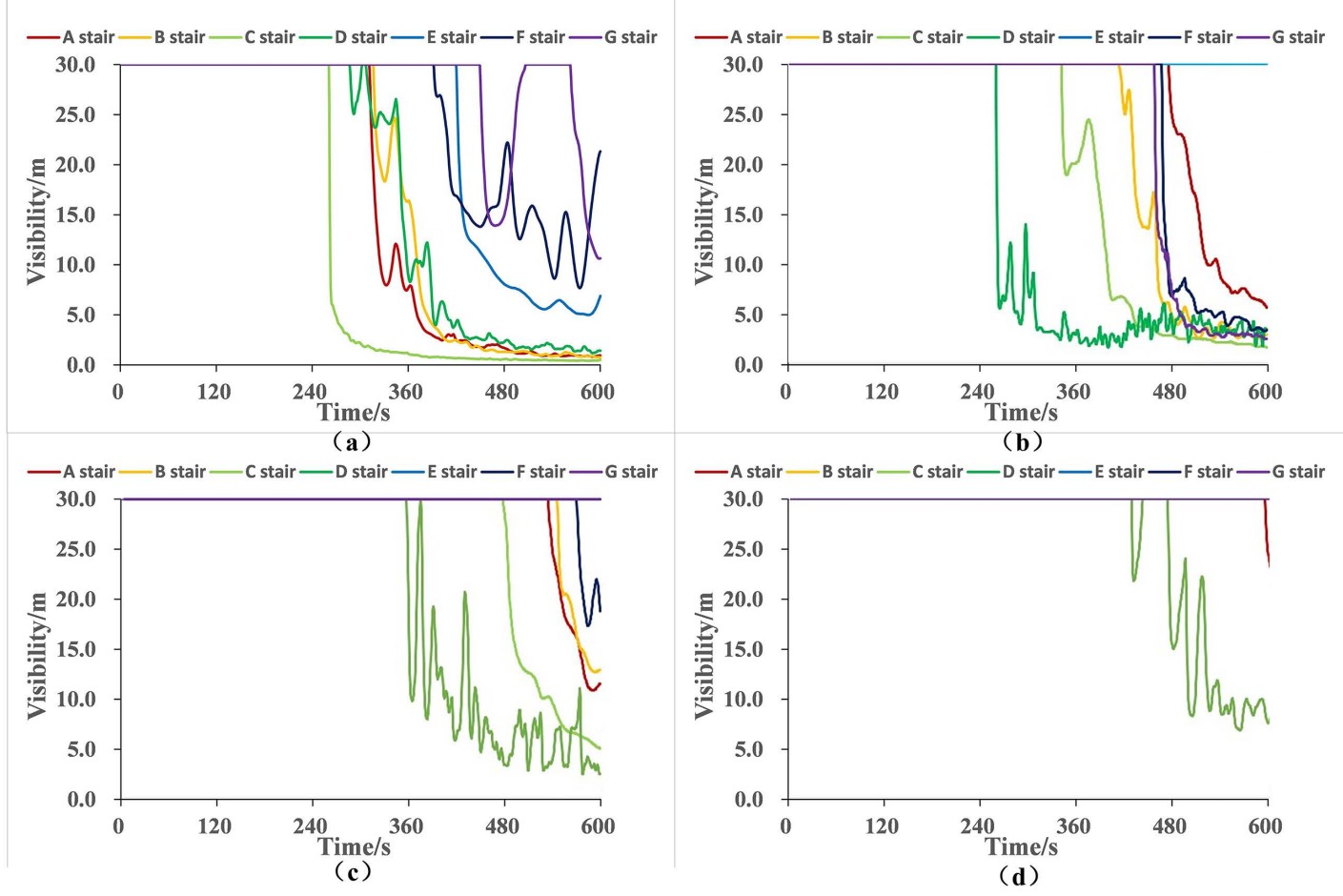

**Fig 11. Simulation Results of Visibility Changes at Stairway Entrances on Each Floor.**

critical value of 10 m, and the required safe evacuation time is determined by the time used for personnel evacuation through each staircase.

By comparing and calculating the differences between the available safe evacuation time and the required safe evacuation time of each staircase, the remaining available evacuation time of each staircase is obtained. The more remaining available evacuation time, the larger the safety margin and the stronger the safety of the staircase. The stairway entrances with different safety margins are classified from long to short in terms of time as: excellent safety, good safety, qualified safety, and safety risk.

As can be seen from Table 4, the fourth – floor of the shopping mall is the farthest from the fire source on the first floor and has the fewest number of people distributed on this floor. It takes more time for smoke to spread to this floor, and personnel can use different staircases to evacuate downstairs more quickly and conveniently. Among the stairway entrances on the fourth floor, only the D stairway entrance has a significantly shorter available safe evacuation time of 496.2 s due to the diffusion of smoke, while the available safe evacuation times of the other stairway entrances are all greater than 600 s. The required safe evacuation times of all stairway entrances on the fourth floor are less than those on other floors, which means that it takes less time for the crowd to evacuate from this floor. Under the combined influence of these two factors, the safety margins on the fourth floor are all greater than 360 s, and the safety performance is excellent.

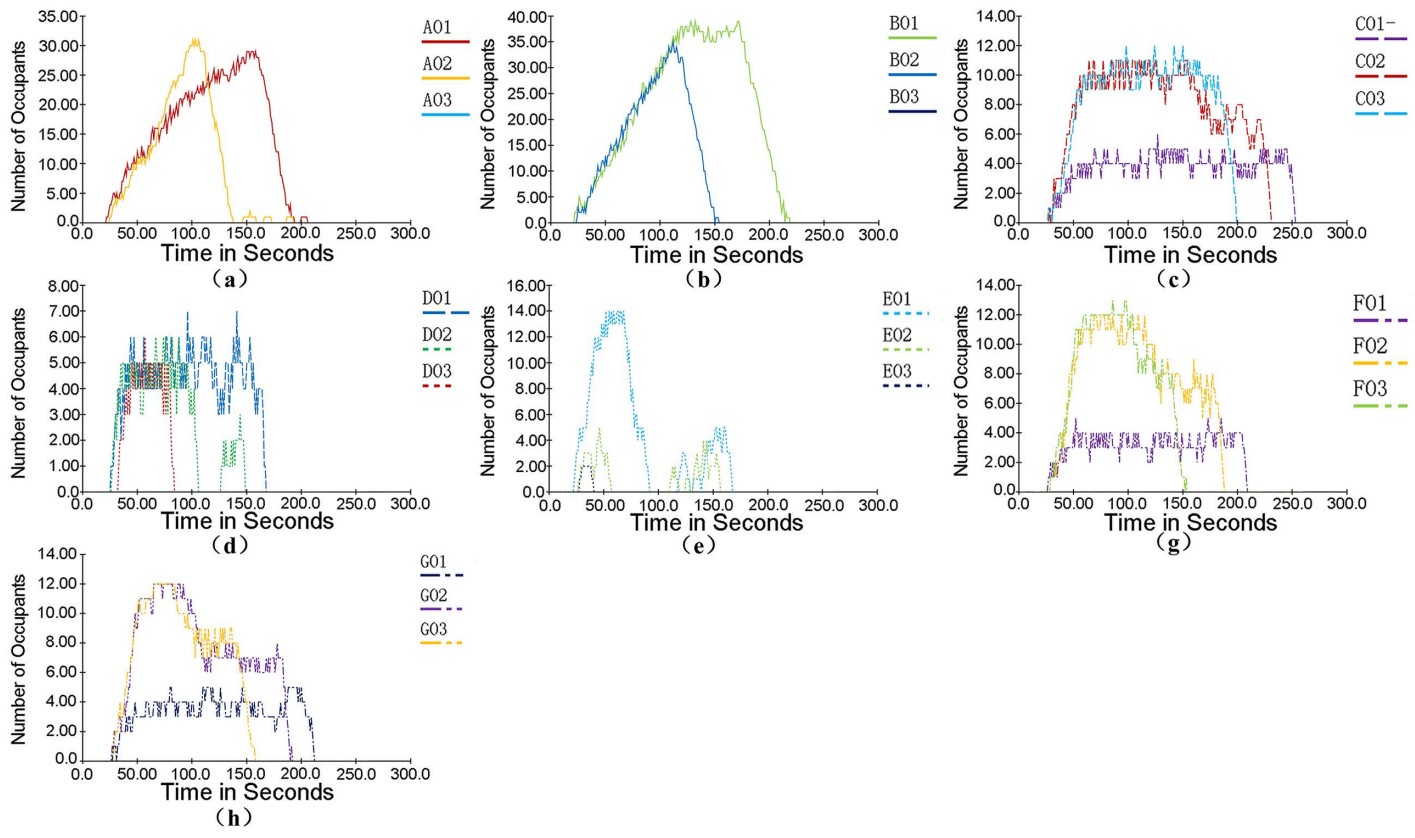

**Fig 12. Usage of Each Staircase for Personnel Evacuation.**

In the third floor of the shopping mall, smoke mainly spreads into the third – floor space through the D stairway. As a result, the available safe evacuation time at the D stairway entrance is significantly shorter than that of the other stairway entrances on this floor, only 361.8 s. There is also a small amount of smoke diffusion at A, B, and C, but only the visibility at the C stairway entrance drops significantly below 10 m, making the available safe evacuation time at the C stairway entrance 526.9 s, which is less than that of the B and C stairway entrances. The other stairway entrances do not show obvious smoke diffusion. Personnel on the third floor mainly evacuate downstairs through the A, B, C, F, and G stairway entrances. The longer time used by personnel during the evacuation process through these stairways leads to an increase in the required safe evacuation time, resulting in a decrease in the safety margin and relatively good safety performance of the A, B, C, and D stairway entrances. The available safe evacuation times at the E, F, and G stairway entrances are longer, and the safety performance is excellent.

The second floor is the closest to the fire source on the first floor and is the most vulnerable to smoke influence among all floors except the first floor. Smoke first spreads into the second floor through the C and D stairway entrances, causing the visibility nearby to drop rapidly. Among them, the available safe evacuation time at the D stairway entrance is the shortest, only 261.6 s, and the safety margin is 85.6 s, posing a safety risk. Although more people pass through the C stairway entrance and the required safe evacuation time is longer, due to the fact that smoke mainly spreads through the D stairway, its available safe evacuation time is longer than that of the D stairway entrance, with a safety margin of 123.4 s and qualified safety. The A, B, F, and G stairway entrances are farther from the main smoke – spreading area (D stairway entrance) on the second floor. There are more rooms in between to slow down the spread of smoke, resulting in a larger

**Table 4. Fire evacuation safety performance analysis results for the shopping mall.**

| Floor | Staircase | Available safety egress time (s) | Required safe evacuation time (s) | Safety margin (s) | Safety performance |
|---|---|---|---|---|---|
| First floor | A | 325.2 | 243.2 | 82 | Security risk |
| | B | 371.4 | 215.6 | 155.8 | Qualified |
| | C | 262.9 | 299.6 | −36.7 | Security risk |
| | D | 358.8 | 198.8 | 160 | Qualified |
| | E | 461.5 | 198.8 | 262.7 | Good |
| | F | 537.6 | 246.8 | 290.8 | Good |
| | G | 600 | 250.4 | 349.6 | Good |
| Second floor | A | 524.5 | 227.6 | 296.9 | Good |
| | B | 462.7 | 161.6 | 301.1 | Good |
| | C | 399 | 275.6 | 123.4 | Qualified |
| | D | 261.6 | 176 | 85.6 | Security risk |
| | E | 600 | 184.4 | 415.6 | Excellent |
| | F | 475.3 | 222.8 | 252.5 | Good |
| | G | 476.4 | 226.4 | 250 | Good |
| Third floor | A | 600 | 326.6 | 273.4 | Good |
| | B | 600 | 326.6 | 273.4 | Good |
| | C | 526.9 | 236 | 290.9 | Good |
| | D | 361.8 | 96.8 | 265 | Good |
| | E | 600 | 96.8 | 503.2 | Excellent |
| | F | 600 | 182 | 418 | Excellent |
| | G | 600 | 185.6 | 414.4 | Excellent |
| Fourth floor | A | 600 | 67.64 | 532.36 | Excellent |
| | B | 600 | 69.68 | 530.32 | Excellent |
| | C | 600 | 156.8 | 443.2 | Excellent |
| | D | 496.2 | 81.8 | 414.4 | Excellent |
| | E | 600 | 66.44 | 533.56 | Excellent |
| | F | 600 | 127.4 | 472.6 | Excellent |
| | G | 600 | 132.44 | 467.56 | Excellent |

safety margin and good safety. The E stairway entrance adopts a different staircase structure from the first floor on the second floor, with stronger personnel evacuation capacity and the ability to slow down the spread of smoke into the stairwell, so its safety performance is excellent.

Since the fire source is located in the restaurant area on the first floor, smoke spreads rapidly on the first floor, causing an increase in the concentration of toxic gases, an increase in smoke temperature, and a decrease in visibility. Moreover, the crowd evacuating from other floors all move downstairs through the staircases, resulting in a longer time required for the staircases on the first floor to complete the personnel evacuation, making the safety margin on the first floor the smallest and posing a relatively large safety risk. The available safe evacuation times at the A and C stairway entrances on the first floor are 325.2 s and 262.9 s respectively, which are the shortest among all stairway entrances. And the required safe evacuation times are relatively long among the stairway entrances on the first floor, being 243.2 s and 299.6 s respectively. This leads to a low safety margin at the A and C stairway entrances on the first floor, posing a safety risk. Among them, the safety of the C stairway entrance is the worst and even insufficient to complete the safe evacuation of personnel, which urgently needs improvement. Compared with the E, F, and G stairway entrances, the A and D stairway entrances are closer to the fire source, resulting in a shorter available safe evacuation time and qualified safety. The safety

performances of the E, F, and G stairway entrances are all good. Among them, the G stairway entrance has the longest available safe evacuation time and the largest safety margin on the first floor because it is close to the exit of the first floor of the shopping mall and the nearby space is large, making it difficult for smoke to stay near the stairway entrance.

## Safety performance optimization

Through the fire simulation digital model, it is evident that there are safety risks at the A and C stairway entrances on the first floor and the D stairway entrance on the second floor of the shopping mall, and there are relatively large differences in safety margins among different stairway entrances on the first and second floors. Therefore, by guiding the evacuating crowd through staircases with higher safety margins, the congestion situation at some staircases can be alleviated, making the personnel distribution more balanced and reasonable, and thus improving the safety margin. The crowd that originally evacuated downstairs through the A, C, and D stairways on the second and third floors is guided to the B, E, F, and G stairway entrances through safety guidance, alleviating the personnel congestion situation and improving the evacuation efficiency.

As can be seen from Fig 13, after the optimization of the personnel evacuation plan, the evacuation time of personnel on the fourth floor remains basically unchanged. The phenomenon of personnel crowding at a few stairway entrances on the second and third floors is reduced, and the overall evacuation speed of personnel is improved. The evacuation time is reduced by 19 s and 16 s respectively. The overall evacuation time of personnel in the shopping mall is 245.5 s, which is reduced by 14.9 s.

The safety margins of stairways A and C (1st floor) and D (2nd floor)—key high-risk areas—were significantly improved. As can be seen from Table 5, the safety margin at the A stairway entrance on the first floor was increased from 82 s to 212.8 s, an increase of 130.8 s. The safety performance has changed from having a safety risk to being qualified. The safety margin at the C stairway entrance on the first floor was increased from less than 0 (insufficient to complete the safe evacuation of personnel) to 78.5 s, which can complete the evacuation of personnel, an increase of 115.2 s. The safety margin at the D stairway entrance on the second floor was increased from 85.6 s to 157.6 s, an increase of 72 s. The safety performance has changed from having a safety risk to being qualified. However, due to the increase in the number of people passing through the other stairway entrances on the first and second floors, the required safe evacuation time has increased, resulting in a certain degree of decrease in their safety margins. But their overall safety margins were still

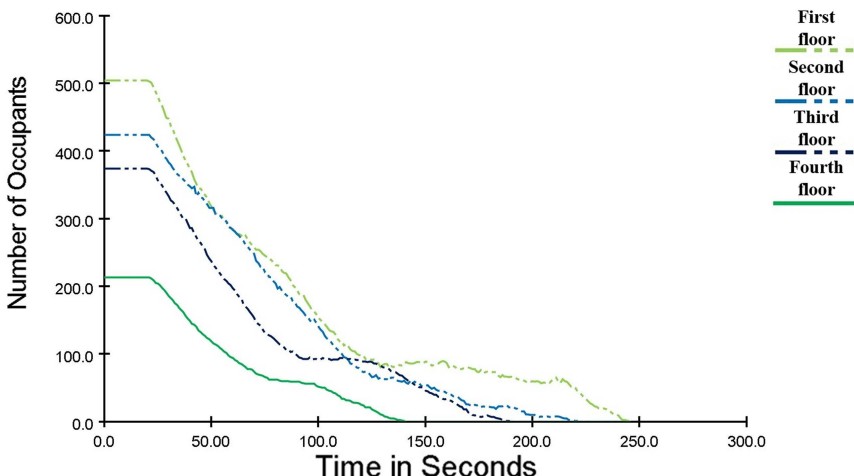

**Fig 13. Curve of the Number of Evacuees on Each Floor of a Shopping Mall Changing with Time after Optimization.**

**Table 5. Analysis results of safety performance of each staircase on the first and second floors of a shopping mall after optimization of personnel evacuation plan.**

| Floor | Staircase | Safety margin before optimization (s) | Optimized safety margin (s) | Safety performance |
|---|---|---|---|---|
| First floor | A | 82 | 212.8 | Qualified |
| | B | 155.8 | 106.6 | Qualified |
| | C | −36.7 | 78.5 | Security risk |
| | D | 160 | 152.8 | Qualified |
| | E | 262.7 | 184.7 | Qualified |
| | F | 290.8 | 244 | Good |
| | G | 349.6 | 328 | Good |
| Second floor | A | 296.9 | 463.7 | Excellent |
| | B | 301.1 | 277.1 | Good |
| | C | 123.4 | 226.6 | Qualified |
| | D | 85.6 | 157.6 | Qualified |
| | E | 415.6 | 367.6 | Excellent |
| | F | 252.5 | 204.5 | Qualified |
| | G | 250 | 233.2 | Qualified |

above the qualified safety performance level. Moreover, the average safety margins on the first and second floors were increased from 180.6 s and 246.4 s to 186.8 s and 275.8 s respectively, and the overall safety margin has increased. As shown in Figs 14 and 15, from the perspective of the overall evacuation on the first and second floors, the differences in safety margins among various stairway entrances narrowed after the improvement, indicating that the safety performance across different stairway entrances became more balanced. This ensures that crowds on different evacuation routes can all evacuate safely, providing more options for people in terms of evacuation paths.

As can be seen from Fig 16, after the optimization of the personnel evacuation scheme, the pedestrian flow rates at exits with larger safety margins, such as Exit 1, Exit 2, and Exit 6, have significantly increased compared to those before optimization. The maximum pedestrian flow rate at Exit 2 reaches 4 people/s and remains above 2 people/s throughout the process, while that at Exit 6 stays above 1.5 people/s. This indicates an improved utilization rate of these exits, an accelerated personnel evacuation speed, reduced crowd congestion, and enhanced evacuation safety.

For exits in areas more affected by the fire with lower safety margins, including Exit 3, Exit 5, and Exit 7, the number of evacuees has been reduced by guiding people to other exits, thereby avoiding congestion caused by excessive crowd gathering. Specifically, the required safe evacuation time at Exit 5 has decreased from 260.4 s to 217.58 s, and that at Exit 7 has dropped from 233.63 s to 134.74 s. In contrast, the required safe evacuation time at Exit 3 has changed slightly, with a reduction of only 4.15 s, which helps improve its safety margin. As for Exit 4, due to the small number of people passing through, there is little difference in its performance before and after the optimization.

## Conclusion

This study proposes a fire numerical simulation method for large-scale urban complexes and optimizes evacuation plans to enhance personnel safety and evacuation efficiency. Taking a shopping mall in Chengdu as an example, fire evacuation simulation is carried out. By analyzing the fire spread situation and comparing the safety performance of stairway entrances on different floors, the evacuation plan is optimized. The overall evacuation time before and after optimization and the safety performance of key areas are analyzed, and the following conclusions are drawn:

(1) Visibility has a more rapid and direct impact on the evacuation process compared to smoke temperature and CO concentration. When smoke spreads, the decrease in visibility has an immediate impact on personnel evacuation.

   

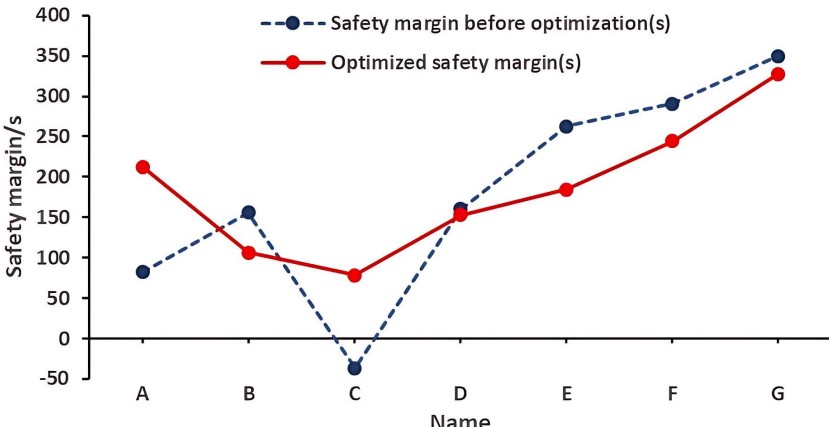

**Fig 14. Safety Margins of Each Staircase on the First Floor after Optimization of Personnel Evacuation Plan.**

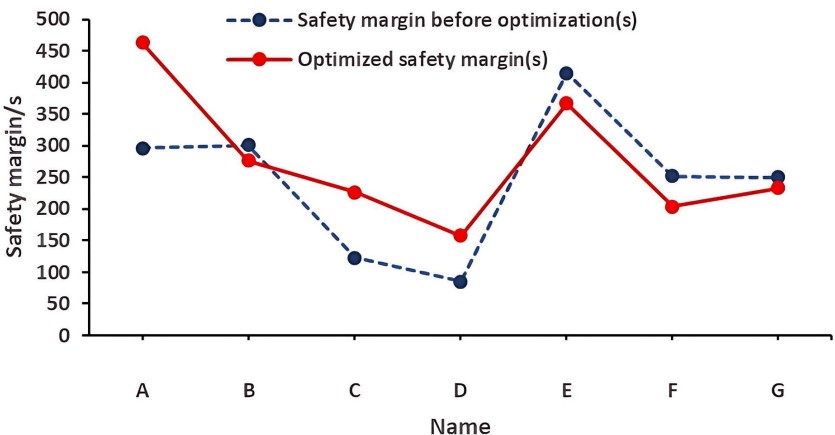

**Fig 15. Safety Margins of Each Staircase on the Second Floor after Optimization of Personnel Evacuation Plan.**

Although the increase in smoke temperature and CO concentration can also pose a serious threat to the personal safety of evacuees, it takes a certain amount of time for the increase in smoke temperature and CO concentration to reach a lethal level for the human body. Therefore, in the fire simulation process, the influence range and degree of visibility are greater than those of smoke temperature and CO concentration.

(2) By comparing the available safe evacuation time and required safe evacuation time for personnel evacuation at staircases across different floors, as well as the pedestrian flow rates and required safe evacuation time among various exits, the safety margins of each staircase were quantified. Furthermore, adjustments were made to personnel evacuation routes and directions based on the fire-affected area. This approach can more intuitively reflect the differences in safety performance among various components, thereby facilitating the proposal and implementation of reasonable optimization schemes. Focusing on staircases A, C, and D with potential safety risks in the shopping mall, as well as exits significantly affected by the fire, a comparison of the fire evacuation safety performance of the shopping mall before and after the optimization of fire protection design was conducted. Simulation results show that the safety

 

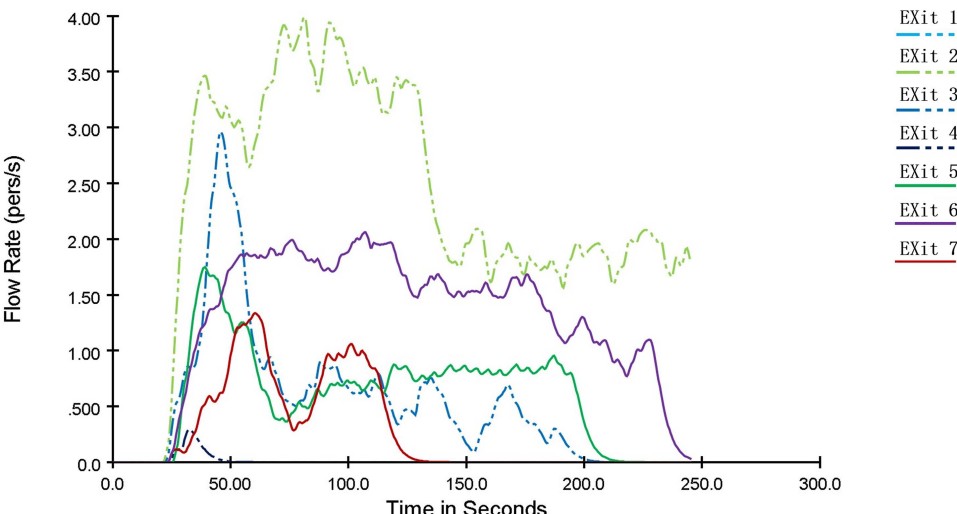

**Fig 16. Curve diagram of the variation of pedestrian flow rate with time at each exit after optimization of the personnel evacuation scheme.**

margins of staircase A and C on the first floor, and staircase D on the second floor increased by 130.8 s, 115.2 s, and 72 s, respectively. Additionally, the overall personnel evacuation time has been reduced by 14.9 s, the distribution of evacuees has been significantly optimized, and the overall evacuation safety has been improved.

(3) Based on fire evacuation simulation data and personnel evacuation simulation data, optimizing the personnel evacuation scheme can enhance the fire evacuation safety performance and improve the evacuation efficiency of the shopping mall. The research results of this paper can provide technical and methodological support for fire safety performance assessment and personnel evacuation scheme planning of large urban complexes.

## Author contributions

**Conceptualization:** Yunhao Jiang, Gang Liu.

**Data curation:** Yulun Du.

**Formal analysis:** Yunhao Jiang, Gang Liu, Siteng Cai, Xiangbing Zhou.

**Funding acquisition:** Gang Liu.

**Investigation:** Yunhao Jiang, Yulun Du.

**Methodology:** Yunhao Jiang, Gang Liu, Xiangbing Zhou.

**Project administration:** Gang Liu, Jing He, Xiangbing Zhou.

**Software:** Siteng Cai, Jing He.

**Supervision:** Gang Liu, Jing He, Xiangbing Zhou.

**Validation:** Yunhao Jiang, Yulun Du, Zhichao Si.

**Visualization:** Yunhao Jiang, Yulun Du.

**Writing – original draft:** Yunhao Jiang.

**Writing – review & editing:** Gang Liu, Jing He, Xiangbing Zhou.

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
