## [Decision Letter · Decision Letter 0]

26 Jun 2025

Dear Dr. Liu,

Thank you for submitting your manuscript to PLOS ONE. After careful consideration, we feel that it has merit but does not fully meet PLOS ONE’s publication criteria as it currently stands. Therefore, we invite you to submit a revised version of the manuscript that addresses the points raised during the review process.

We look forward to receiving your revised manuscript.

Kind regards,

Gianluca Genovese, Ph.D.

Academic Editor

PLOS ONE

 [Open Funding of Sichuan Provincial Key Laboratory of Philosophy and Social Sciences for Mountain Tourism Safety (Grant number 24SDLYAQZZ001), National Natural Science Foundation of China Project (Grant number 42371418)]. 

Reviewers' comments:

Reviewer's Responses to Questions

**Comments to the Author**

1. Is the manuscript technically sound, and do the data support the conclusions?

Reviewer #1: Partly

Reviewer #2: Yes

2. Has the statistical analysis been performed appropriately and rigorously?

Reviewer #1: No

Reviewer #2: Yes

3. Have the authors made all data underlying the findings in their manuscript fully available?

Reviewer #1: No

Reviewer #2: Yes

4. Is the manuscript presented in an intelligible fashion and written in standard English?

Reviewer #1: No

Reviewer #2: Yes

Reviewer #1: This paper presents a research on analysis and optimization of fire evacuation safety performance in large urban complexes based on the Building Information Modeling (BIM). However, the description of the abstract needs to be redesigned.

What is the most important innovation point in this paper?

The description of data analysis is not detailed enough.

It is noted that your manuscript needs careful editing by someone withexpertise in technical English editing paying particular attention to Englishgrammar, spelling, and sentence structure so that the goals and results of thestudy are clear to the reader.

Figures 6, 9, and 10 are not clear.

It lacks grid independence verification in numerical simulation calculations.

The description of material properties parameters in simulation calculations is incomplete.

Figure 3 illustrates the temperature changes across different floors of the building. There has been no in-depth analysis of the reasons behind these phenomena.

The conclusion section is not clear enough, it is best to write three separate conclusions.

Reviewer #2: The main title is develop very well. The main idea is develop by chronological order. The main idea is develops very well. The main idea is discuss about STEM. The idea of BIM is develop very well. It should elaborate more.

**Do you want your identity to be public for this peer review?** For information about this choice, including consent withdrawal, please see our Privacy Policy

Reviewer #1: No

Reviewer #2: **Yes**

---

## [Author Response · Author response to Decision Letter 1]

31 Jul 2025

Dear Editor:

Manuscript ID PONE-D-25-24987 entitled " Analysis and Optimization of Fire Evacuation Safety Performance in Large Urban Complexes " which has been revised.

We are truly grateful to yours and the reviewers’ critical comments and thoughtful suggestions. Based on these comments and suggestions, we have made significant modifications on the original manuscript. All changes made to the text are in red color.

We hope the new manuscript will meet your magazine’s standard. Please do forward our heartfelt thanks to these experts. We look forward to hearing from you soon for a favorable decision.

We tried our best to improve the manuscript and made a significant change in the manuscript. Below you will find our point-by-point responses to the reviewers’ and editor’s comments/ questions.

Reviewers' comments:

Reviewer #1:

This paper presents a research on analysis and optimization of fire evacuation safety performance in large urban complexes based on the Building Information Modeling (BIM).

1. However, the description of the abstract needs to be redesigned.

Reply: Many thanks for this comment. After review and proofreading, we have comprehensively revised the introduction section in accordance with your suggestions, with improvements made to its logical structure. The narrative logic of the introduction has been reorganized, adopting the logical thread of "research background—existing problems—research objectives—research methods—case verification—research significance". The feasibility and effectiveness of the research have been demonstrated through specific methods and case data.

The following is the revised content:

“

Urban large-scale complexes, such as shopping malls, pose significant challenges for fire safety management due to their intricate spatial layouts, high population density, and diverse occupancy characteristics. Efficient fire evacuation strategies are critical to minimizing casualties and economic losses, yet existing approaches often overlook the dynamic interplay between fire propagation and human behavior, leading to suboptimal safety assessments. This study proposes an integrated simulation framework to optimize evacuation strategies by coupling fire dynamics with pedestrian flow modeling, aiming to enhance both evacuation efficiency and personnel safety. The methodology comprises three key steps: (1) Fire scenario simulation: A Building Information Modeling (BIM)-based digital platform is constructed to simulate fire propagation. Critical fire parameters (e.g., heat release rate, combustion model) are calibrated to quantify temporal variations in smoke temperature, CO concentration, and visibility across different zones. (2) Evacuation dynamics modeling: A pedestrian evacuation model is developed by integrating demographic factors (age structure, movement speed, population density) and fire-induced regional risks, enabling realistic simulation of crowd movement under fire conditions. (3) Safety performance evaluation and strategy optimization: Safety margins at staircases are assessed by comparing Required Safe Egress Time (RSET) and Available Safe Egress Time (ASET), followed by a safety grading system to identify high-risk bottlenecks. Evacuation strategies are then optimized to mitigate these risks. A case study was conducted on a shopping mall in Chengdu to validate the framework. Simulation results indicate an initial evacuation time of 260.4 seconds. Safety performance analysis revealed critical risks at staircases A and C (1st floor) and D (2nd floor) due to insufficient safety margins. After strategy optimization, the total evacuation time was reduced to 245.5 seconds, with safety margins at the three high-risk staircases increased by 130.8 s, 115.2 s, and 72 s, respectively, fully meeting safety requirements. The overall evacuation efficiency was significantly improved. This study demonstrates the effectiveness of the proposed framework in quantifying fire risks and optimizing evacuation strategies for large-scale complexes. The integrated simulation approach provides a scientific basis for evidence-based safety management and evacuation planning, offering valuable insights for urban fire safety engineering and emergency response optimization.

”

2. What is the most important innovation point in this paper?

Reply: Many thanks for this comment. We have re-refined and sorted out the innovation points of this paper. The most important innovation of this study is: proposing an integrated simulation framework, which obtains a personnel evacuation model under the influence of fire by coupling fire dynamics with pedestrian flow modeling. Taking this model as the data basis, safety performance evaluation is carried out and evacuation strategies are optimized, aiming to improve both evacuation efficiency and personnel safety. In addition, simulations based on real scenarios are conducted to demonstrate the effectiveness and applicability of this method. We further highlight the innovation in the revised manuscript.

3. The description of data analysis is not detailed enough.

Reply: Many thanks for this comment. We have supplemented the insufficiently detailed parts in the data analysis. These include: the variation patterns of visibility across different floors in the fire simulation results and their causes; in the pedestrian evacuation simulation, further statistics and analysis on personnel's exit choices. By comparing the pedestrian flow rates and usage times among different exits, a more detailed personnel evacuation scenario has been constructed. Additionally, the safety of personnel's exit choices has been incorporated into the optimization analysis of personnel evacuation to enhance the overall safety and rationality of the optimization scheme.

The following is the revised section:

“

Figure 6 Simulation results of visibility variation across floors in the shopping center

Visibility is generally defined as the maximum distance at which a person can clearly see an object. The visibility conditions on each floor are illustrated in Figure 6. When a fire breaks out, a large amount of smoke is generated and quickly spreads to the surrounding rooms. By 142.5 seconds, the visibility in the first-floor restaurant drops below 10 meters, showing obvious horizontal diffusion characteristics. After 374.7 seconds of the fire, the smoke spreads to the vicinity of staircases C and D, and the vertical upward spread of the smoke causes the visibility on the second floor to gradually drop below 10 meters, with the smoke continuing to spread to upper floors. After 571.3 seconds of the fire, due to the deposition of smoke particles under the action of gravity, the visibility in most areas of the first and second floors continues to decrease. The corridor areas where staircases A and B are located have narrow spaces and poor ventilation conditions, leading to severe smoke accumulation and a rapid drop in visibility to below 10 meters. As for staircases C and D, their closer distance to the fire source results in longer smoke accumulation time, causing their visibility to drop below 10 meters. At the end of the 600-second simulation, the visibility in most positions on the first and second floors is below 10 meters, with only the areas near staircases E and G being less affected in terms of visibility. On the third floor, the visibility at staircases A, B, C, and the lobby near staircase D drops below 10 meters, while the visibility in other areas remains good.

Figure 9. Curve diagram of the variation of pedestrian flow rate with time at each exit of a shopping mall

The variation of pedestrian flow rate at each exit with evacuation time is shown in Figure 9. During the evacuation process, people tend to choose the nearest exit as their target. At 50 seconds after the start of evacuation, the pedestrian flow rate at each exit reaches its peak: the flow rate at Exit 1 is the highest, reaching 6.17 people/s; the flow rates at Exit 2 and Exit 5 reach 3 people/s; and the maximum flow rates at the remaining exits are around 2 people/s. Afterwards, the flow rates gradually decrease until 110 seconds into the evacuation, when the pedestrian flow rate at each exit stabilizes at approximately 1 people/s. As indicated in the table, without evacuation guidance, although the pedestrian flow rates at each exit stabilize after a certain period, the distribution of people is uneven and fails to avoid fire-affected areas, resulting in reduced evacuation efficiency and a narrowed safety margin.

Figure 16. Curve diagram of the variation of pedestrian flow rate with time at each exit after optimization of the personnel evacuation scheme

As can be seen from Figure 16, after the optimization of the personnel evacuation scheme, the pedestrian flow rates at exits with larger safety margins, such as Exit 1, Exit 2, and Exit 6, have significantly increased compared to those before optimization. The maximum pedestrian flow rate at Exit 2 reaches 4 people/s and remains above 2 people/s throughout the process, while that at Exit 6 stays above 1.5 people/s. This indicates an improved utilization rate of these exits, an accelerated personnel evacuation speed, reduced crowd congestion, and enhanced evacuation safety.

For exits in areas more affected by the fire with lower safety margins, including Exit 3, Exit 5, and Exit 7, the number of evacuees has been reduced by guiding people to other exits, thereby avoiding congestion caused by excessive crowd gathering. Specifically, the required safe evacuation time at Exit 5 has decreased from 260.53 s to 217.58 s, and that at Exit 7 has dropped from 233.63 s to 134.74 s. In contrast, the required safe evacuation time at Exit 3 has changed slightly, with a reduction of only 4.15 s, which helps improve its safety margin. As for Exit 4, due to the small number of people passing through, there is little difference in its performance before and after the optimization.

”

4. It is noted that your manuscript needs careful editing by someone with expertise in technical English editing paying particular attention to English grammar, spelling, and sentence structure so that the goals and results of the study are clear to the reader.

Reply: Many thanks for this comment. After our re-examination and proofreading, targeting the issues of grammar, tense, sentence structure, and spelling in the original text, we have corrected the incorrect spellings and sentence structures, unified the tenses, and improved the grammar to make it more consistent with the usage conventions of academic English.

5. Figures 6, 9, and 10 are not clear.

Reply: Many thanks for this comment. We have redrawn the aforementioned images with insufficient clarity, increased the text size in the images, and improved their quality and resolution. Figures 7, 11, and 12 in the revised manuscript correspond to Figures 6, 9, and 10 in the original manuscript.

The redrawn picture is as follows:

Figure 7. Curve of the Number of Evacuees on Each Floor of a Shopping Mall Changing with Time

Figure 11. Simulation Results of Visibility Changes at Stairway Entrances on Each Floor

Figure 12. Usage of Each Staircase for Personnel Evacuation

6. It lacks grid independence verification in numerical simulation calculations.

Reply: Many thanks for this comment. In the section "Fire Simulation Parameter Settings", we have added content related to grid independence verification. Specifically, by comparing the variation trends of fire products near the fire source under different grid size conditions, a consistency test was conducted to analyze the degree of differences, thereby performing the grid independence verification.

The following is the revised section:

“……

To verify grid independence, the following criteria are typically required:

1. Grid Convergence: When calculations are performed using different grids, the results should converge to similar solutions under sufficiently small grid sizes. As the grid size decreases, the computational results should gradually approach a stable solution.

2. Grid Independence: When calculations are conducted using different grids, the results should remain consistent under the same physical conditions. Even with varying grid sizes, the computational results should exhibit the same physical trends and characteristics.

3. Grid Smoothness: When calculations are carried out using different grids, the results should remain smooth under the same physical conditions. Even as the grid size changes, the computational results should not display significant isolated or singular points.

Therefore, in this paper, three groups of progressively decreasing grid sizes (1.2 m × 1.2 m × 1.2 m, 0.6 m × 0.6 m × 0.6 m, and 0.2 m × 0.2 m × 0.2 m) were set, and the variations in key indicators such as temperature, visibility, and CO concentration distribution near the fire source were recorded. By comparing the differences in the variation curves of the yield of the same fire product under different grid scales and conducting a significance analysis of these differences, the grid independence is verified.

Figure 2 Variation of Fire Product Yields Under Different Grid Sizes

As indicated in Figure 2, and the variation trends are similar without significant divergence points. In the significance analysis of differences, all P-values are greater than 0.05, indicating no significant difference in the mean values of the data. Therefore, grid independence is verified.

……”

7. The description of material properties parameters in simulation calculations is incomplete.

Reply: Many thanks for this comment. We have supplemented the incomplete parts of the material property parameters in the simulation calculations. The supplemented material coefficients include material type, density, specific heat, electrical conductivity, emissivity, absorption coefficient.

The following is the revised section:

“

The heat release rate growth coefficient depends on the type of combustible material, and different types of combustibles can lead to distinct types of fires. Shopping malls contain a large variety of combustible materials, such as wood, PVC, and foam materials that are widely used in decoration processes. In this paper, the types of combustible materials are set as wood, PVC, and foam materials, while the walls are set as concrete. The material parameters are presented in Table 1.

Table 1 Material Attribute Parameter Table

Material Type Density Specific Heat Conductivity Emissivity Absorption Coefficient

Wood 640 kg/m³ 2.85 kJ/(kg·K) 0.14 W/(m·K) 0.9 5.0E+4 1/m

PVC 1380 kg/m³ 1.44 kJ/(kg·K) 0.16 W/(m·K) 0.95 5.0E+4 1/m

Foam 28 kg/m³ 1.7 kJ/(kg·K) 0.05 W/(m·K) 0.9 5.0E+4 1/m

Concrete 2280 kg/m³ 1.04 kJ/(kg·K) 1.8 W/(m·K) 0.9 5.0E+4 1/m

”

8. Figure 3 illustrates the temperature changes across different floors of the building. There has been no in-depth analysis of the reasons behind these phenomena.

Reply: Many thanks for this comment. We have re-analyzed this part and summarized the variation law of fire temperature across different floors, with supplementary content added to the original analysis.

The following is the revised section:

“

Figure 4 Simulation results of smoke temperature variation across floors in the shopping center

Figure 4 illustrates the temperature changes across different floors of the building. At 161 s after fire ignition, the smoke temperature in the first-floor restaurant reached 60°C. By the end of the simulation (600 s), only areas such as stairwells E and G on the first floor approached 60°C, while temperatures on other floors remained below this critical threshold.

As time elapses after ignition, due to the narrow space and limited airflow in the room where the fire source is located, heat accumulates more easily within the space, resulting in significant temperature changes. In contrast, in other areas, being relatively far from the fire source, and after fire smoke enters the hall, the internal space of the hall is larger, heat exchange is frequent, and the smoke dissipates more easily. These areas are significantly affected by the attenuation of thermal radiation and smoke energy transfer, leading to smaller temperature variations, all of which remain below 60°C.

”

---

## [Decision Letter · Decision Letter 1]

3 Sep 2025

Dear Dr. Liu,

Thank you for submitting your manuscript to PLOS ONE. After careful consideration, we feel that it has merit but does not fully meet PLOS ONE’s publication criteria as it currently stands. Therefore, we invite you to submit a revised version of the manuscript that addresses the points raised during the review process.

We look forward to receiving your revised manuscript.

Kind regards,

Peng Li, Ph.D.

Academic Editor

PLOS ONE

Journal Requirements:

**Additional Editor Comments:**

Reviewer #1: All comments have been addressed.

Reviewer #3:The authors have thoroughly addressed the reviewers' concerns. The addition of grid independence verification, complete material property parameters, and enhanced analysis of evacuation dynamics significantly strengthen the manuscript. The integrated BIM-based framework provides practical value for fire safety assessment in large commercial complexes.

Minor Suggestions for Final Preparation

While I recommend acceptance, the authors may wish to address these minor points during final manuscript preparation:

1. Numerical Consistency: Please verify the evacuation time at Exit 5 (appears as both 260.4s and 260.53s in different sections). Also make the height unit consistent, choose m or cm.

2. Statistical Test Specification: When stating "all P-values are greater than 0.05" for grid independence verification, please specify which statistical test was used.

3. Figure 2: Consider adding brief labels to panels (a), (b), (c) to indicate which parameter each represents.

4. Minor Typographical: "be-tween" appears with incorrect hyphenation in one location.

These are minor editorial points that can be easily addressed during production. The scientific content and methodology are sound, and the manuscript makes a valuable contribution to fire evacuation safety research.

Reviewers' comments:

Reviewer's Responses to Questions

**Comments to the Author**

Reviewer #1: All comments have been addressed

Reviewer #3: All comments have been addressed

2. Is the manuscript technically sound, and do the data support the conclusions?

Reviewer #1: Yes

Reviewer #3: Yes

3. Has the statistical analysis been performed appropriately and rigorously?

Reviewer #1: N/A

Reviewer #3: Yes

4. Have the authors made all data underlying the findings in their manuscript fully available?

Reviewer #1: Yes

Reviewer #3: Yes

5. Is the manuscript presented in an intelligible fashion and written in standard English?

Reviewer #1: Yes

Reviewer #3: Yes

Reviewer #1: (No Response)

Reviewer #3: The authors have thoroughly addressed the reviewers' concerns. The addition of grid independence verification, complete material property parameters, and enhanced analysis of evacuation dynamics significantly strengthen the manuscript. The integrated BIM-based framework provides practical value for fire safety assessment in large commercial complexes.

Minor Suggestions for Final Preparation

While I recommend acceptance, the authors may wish to address these minor points during final manuscript preparation:

1. Numerical Consistency: Please verify the evacuation time at Exit 5 (appears as both 260.4s and 260.53s in different sections). Also make the height unit consistent, choose m or cm.

2. Statistical Test Specification: When stating "all P-values are greater than 0.05" for grid independence verification, please specify which statistical test was used.

3. Figure 2: Consider adding brief labels to panels (a), (b), (c) to indicate which parameter each represents.

4. Minor Typographical: "be-tween" appears with incorrect hyphenation in one location.

These are minor editorial points that can be easily addressed during production. The scientific content and methodology are sound, and the manuscript makes a valuable contribution to fire evacuation safety research.

**Do you want your identity to be public for this peer review?** For information about this choice, including consent withdrawal, please see our Privacy Policy

Reviewer #1: No

Reviewer #3: No

---

## [Author Response · Author response to Decision Letter 2]

21 Sep 2025

Dear Editor:

Manuscript ID PONE-D-25-24987R1 entitled "Analysis and Optimization of Fire Evacuation Safety Performance in Large Urban Complexes" which has been revised.

We are truly grateful to yours and the reviewers’ critical comments and thoughtful suggestions. Based on these comments and suggestions, we have made significant modifications on the original manuscript. All changes made to the text are in red color in the “Revised Manuscript with Track Changes” document.

We hope the new manuscript will meet your magazine’s standard. Please do forward our heartfelt thanks to these experts. We look forward to hearing from you soon for a favorable decision.

We tried our best to improve the manuscript and made a significant change in the manuscript. Below you will find our point-by-point responses to the reviewers’ and editor’s comments/ questions.

#Reviewers' comments:

The authors have thoroughly addressed the reviewers' concerns. The addition of grid independence verification, complete material property parameters, and enhanced analysis of evacuation dynamics significantly strengthen the manuscript. The integrated BIM-based framework provides practical value for fire safety assessment in large commercial complexes.

Minor Suggestions for Final Preparation

While I recommend acceptance, the authors may wish to address these minor points during final manuscript preparation:

1. Numerical Consistency: Please verify the evacuation time at Exit 5 (appears as both 260.4s and 260.53s in different sections). Also make the height unit consistent, choose m or cm.

Reply: Many thanks for this comment. Regarding the issue of inconsistencies in some experimental results, we re-examined the experimental results and unified the experimental conclusions across the study. Additionally, the unit for height was standardized to meters (m).

2. Statistical Test Specification: When stating "all P-values are greater than 0.05" for grid independence verification, please specify which statistical test was used.

Reply: Many thanks for this comment. To address the issue of ambiguity in the method used for verifying grid independence, we have further elaborated on the applied method—One-way Analysis of Variance (One-Way ANOVA)—and additionally included the calculation process to facilitate a more robust verification of grid independence.

The following is the revised section:

“One-way Analysis of Variance (One-Way ANOVA) is a statistical method used to determine whether there are significant differences in the means among multiple groups. Its core principle involves decomposing the total variation into between-group variation and within-group variation, and then using the F-test and P-value to judge the significance of between-group differences. This method is well-suited for the research design of this study, where "grid scale serves as the sole independent variable and key indicators act as continuous dependent variables". It complies with the assumptions of independence, normality, and homogeneity of variance for grid monitoring data, enabling quantitative verification of whether grid scale exerts a significant impact on key indicators and thus accurately aligning with the objective of verifying grid independence.

Therefore, this study set three groups of gradually reduced grid sizes (1.2m×1.2m×1.2m, 0.6m×0.6m×0.6m, and 0.2m×0.2m×0.2m). Taking visibility (V), CO concentration (CO), and temperature (T) near the fire source as the key observation indicators, a one-way ANOVA was conducted based on the continuous monitoring data obtained 600 seconds after the fire ignition (with a sample size of 1001 for each group). The differences in the variation curves of the same fire product yield under different grid scales were compared to verify grid independence. The variation of fire product yields under different grid sizes is illustrated in Figure 2, and the results of the one-way ANOVA are presented in Table 2.

Table 2. One-way ANOVA Results Table of Key Indicators Under Different Grid Scales

Analytical Indicator Grid Scale Group Mean (± Standard Deviation) F-statistic P-value Levene's Test Conclusion (α=0.05)

Visibility (V) 1.2m 10.23±12.65 1.555 0.211 F=1.124

P=0.326 Homogeneous variance;

No significant difference between groups

0.6m 10.87±12.71

0.2m 11.02±12.83

CO Concentration�CO� 1.2m 0.008±0.003 1.624 0.197 F=0.987

P=0.373 Homogeneous variance;

No significant difference between groups

0.6m 0.007±0.002

0.2m 0.007±0.002

Temperature (T) 1.2m 125.68±89.32 0.482 0.618 F=1.352

P=0.259 Homogeneous variance;

No significant difference between groups

0.6m 126.15±88.97

0.2m 127.03±89.15

As shown in Figure 2, the variation trends of different fire product yields with fire development are similar under different grids and there are no significant divergence points. As presented in Table 2, in the results of one - way analysis of variance, among the yields of each fire product under different grid scales, the F - statistics of all indicators are relatively small, and all P - values are greater than 0.05, which meets the homogeneity of variance test. This indicates that changes in grid size will not significantly alter the results of key indicators such as visibility, CO concentration, and temperature, verifying that grid independence holds.”

3. Figure 2: Consider adding brief labels to panels (a), (b), (c) to indicate which parameter each represents.

Reply: Many thanks for this comment. Regarding the issue that the subplots in Figure 2 lack labels, we have divided the subplots and added standardized labels to facilitate better differentiation of different parameters.

The revised figure 2 is as following:

4. Minor Typographical: "be-tween" appears with incorrect hyphenation in one location.

Reply: Many thanks for this comment. Regarding the issues of incorrect punctuation usage in the manuscript, we have conducted a comprehensive check and revised all instances where punctuation was improperly used.

---

## [Decision Letter · Decision Letter 2]

4 Nov 2025

Analysis and Optimization of Fire Evacuation Safety Performance in Large Urban Complexes

PONE-D-25-24987R2

Dear Dr. Liu,

We’re pleased to inform you that your manuscript has been judged scientifically suitable for publication and will be formally accepted for publication once it meets all outstanding technical requirements.

Kind regards,

Shrisha Rao, Ph.D.

Academic Editor

PLOS ONE

Additional Editor Comments (optional):

Reviewers' comments:

Reviewer's Responses to Questions

**Comments to the Author**

Reviewer #1: All comments have been addressed

2. Is the manuscript technically sound, and do the data support the conclusions?

Reviewer #1: Yes

3. Has the statistical analysis been performed appropriately and rigorously?

Reviewer #1: N/A

4. Have the authors made all data underlying the findings in their manuscript fully available?

Reviewer #1: Yes

5. Is the manuscript presented in an intelligible fashion and written in standard English?

Reviewer #1: Yes

Reviewer #1: (No Response)

**Do you want your identity to be public for this peer review?** For information about this choice, including consent withdrawal, please see our Privacy Policy

Reviewer #1: No

---

## [Editor Report · Acceptance letter]

PONE-D-25-24987R2

PLOS ONE

Dear Dr. Liu,

I'm pleased to inform you that your manuscript has been deemed suitable for publication in PLOS ONE. Congratulations! Your manuscript is now being handed over to our production team.

Kind regards,

on behalf of

Dr. Shrisha Rao

Academic Editor

PLOS ONE